# Ligand-Gated Ion Channels: Prognostic and Therapeutic Implications for Gliomas

**DOI:** 10.3390/jpm13050853

**Published:** 2023-05-19

**Authors:** Grace Hey, Rohan Rao, Ashley Carter, Akshay Reddy, Daisy Valle, Anjali Patel, Drashti Patel, Brandon Lucke-Wold, Daniel Pomeranz Krummel, Soma Sengupta

**Affiliations:** 1College of Medicine, University of Florida, Gainesville, FL 32610, USA; 2College of Medicine, University of Cincinnati, Cincinnati, OH 45267, USA; 3Eastern Virginia Medical School, Norfolk, VA 23507, USA; 4Department of Neurosurgery, University of Florida, Gainesville, FL 23608, USA; 5Department of Neurology & Rehabilitation Medicine, University of Cincinnati, Cincinnati, OH 45267, USA

**Keywords:** glioma, ligand-gated ion channels

## Abstract

Gliomas are common primary brain malignancies that remain difficult to treat due to their overall aggressiveness and heterogeneity. Although a variety of therapeutic strategies have been employed for the treatment of gliomas, there is increasing evidence that suggests ligand-gated ion channels (LGICs) can serve as a valuable biomarker and diagnostic tool in the pathogenesis of gliomas. Various LGICs, including P2X, SYT16, and PANX2, have the potential to become altered in the pathogenesis of glioma, which can disrupt the homeostatic activity of neurons, microglia, and astrocytes, further exacerbating the symptoms and progression of glioma. Consequently, LGICs, including purinoceptors, glutamate-gated receptors, and Cys-loop receptors, have been targeted in clinical trials for their potential therapeutic benefit in the diagnosis and treatment of gliomas. In this review, we discuss the role of LGICs in the pathogenesis of glioma, including genetic factors and the effect of altered LGIC activity on the biological functioning of neuronal cells. Additionally, we discuss current and emerging investigations regarding the use of LGICs as a clinical target and potential therapeutic for gliomas.

## 1. Introduction

Gliomas are common primary brain malignancies characterized by diffuse infiltrative growth of tumor cells in the preexistent parenchyma of the central nervous system (CNS), the most common of which include astrocytomas, oligodendrogliomas, and ependymomas [1]. Diagnosis of tumor subtypes is made through histology, immunohistochemistry, radiographic appearance, and occasionally localization. The most recent WHO Classification of Tumors of the Central Nervous System, released in 2021, has incorporated molecular/genetic biomarkers. For example, the guidelines suggest using mutations in IDH, the *TERT* promotor, and epidermal growth factor receptor (EGFR) as markers for glioblastoma (GBM) [2]. The aggressiveness and heterogeneity of gliomas complicate treatment, manifesting in high mortality rates. The current standard of care treatment for gliomas is surgical resection followed by adjuvant chemotherapy, as complete resection is frequently limited by proximity to eloquent areas in the brain [3]. A variety of therapeutic approaches have been utilized for the treatment of gliomas, and recent evidence suggests LGICs can be used as a possible biomarker for prognosis in the development of gliomas [4,5,6].

Communication within the nervous system involves presynaptic neurons receiving electrical stimulation from action potentials. If the action potentials reach the threshold (~55 mV), the presynaptic neuron will release a neurotransmitter into the synaptic cleft at the axon terminal. Once in the synaptic cleft, the neurotransmitter can bind to receptors located on the postsynaptic neuron [7]. LGICs are transmembrane receptors that, upon binding of a ligand (i.e., neurotransmitter) in the extracellular domain (ECD), produce a conformational change that opens the channel in the transmembrane domain (TMD) and allow for ions (Na^+^, K^+^, Ca^2+^ or Cl^−^) to pass through the membrane down their electrochemical gradients [8]. Depending on the ion flow and ion involved, the membrane potential can hyperpolarize, an inhibitory response, or depolarize, an excitatory response [7,8,9]. LGICs mediate a rapid synaptic transmission leading to further signal transduction in the central and peripheral nervous system.

There are three classifications for LGICs: ATP/purine-gated channels, ionotropic glutamate receptors, and Cys-loop receptors (Figure 1). Diverse transmembrane morphology (Figure 1) confers each receptor family with unique physiological functions (Table 1). All three superfamilies have both metabotropic (G protein-coupled) and ionotropic activity. However, for this discussion, we will focus on the receptor’s ionotropic activity.

Ionotropic glutamate receptors form tetramers with three receptor subtypes, α-amino-3-hydroxy-5-methyl-4-isoxazolepropionic acid (AMPA), Kainate, and N-methyl-D-aspartate (NMDA), that are activated upon binding of L-Glutamate to its receptor, permitting the influx of mostly Na^+^ and Ca^+^ into the cells. The proposed link between ion influx and tumor growth is that intracellular Ca^+^ mediates the activation of several pro-proliferative signal transduction pathways, such as Akt, ERK/MAP kinase, and PKA [11,12]. In addition to amino acid synthesis, glutamate is critical for the maintenance of homeostasis within the nervous system, providing significant excitatory innervation within the cortex [10,13]. Furthermore, glutamate is a precursor for glutathione, which may contribute to tumor pathogenesis as it is necessary for scavenging reactive oxygen species (ROS) [10]. Stimulation of the glutamate receptor activates focal adhesion kinase (FAK), an enzyme responsible for the regulation of cell motility, growth, and invasion [4]. Dysregulation of FAK in glioma can promote the invasion of malignant cells into normal brain cells [4].

ATP-gated channels or purinoreceptors, called P2X receptors, are assembled in a trimeric structure and open in response to extracellular adenosine 5′-triphosphate (ATP). ATP is a tightly-regulated molecule critical for a variety of intracellular processes, including providing cells with energy [14]. ATP-gated channels have been found to play a role in phosphorylation in astrocytoma cells, and P2RX4 receptor expression has been associated with activated microglia and unique tumor-associated macrophages [15,16].

The last class of LGICs is pentameric assembling Cys-loop receptors that have a disulfide bond in the loop of the extracellular domain (Figure 1). There are four distinct subtypes named after their corresponding ligands: acetylcholine (nACh), serotonin (5-HT_3_), GABA_A_, and glycine. Of the Cys-loop receptors, glycine receptors have been the most studied in relation to gliomas. Interestingly, increased expression of serine and glycine metabolism-related genes correlated with poorer prognoses in glioma [17]. Serine and glycine are considered immunosuppressive metabolites and thus may contribute to the immune evasion of cancer cells [18,19]. Moreover, it has been suggested that GABA_A_ may be implicated in cell proliferation, possibly impacting tumor pathogenesis [10]. For example, in breast cancer cells, increased expression of *GABRA3* contributes to the activation of Akt, thereby enhancing proliferation via the Akt pathway [20].

## 2. Cells Involved in Glioma Pathogenesis

Ligand channels play a role in the pathogenesis of various gliomas and serve as biomarkers in prognosis [21,22]. These channels, including P2X, SYT16, and PANX2, have a unique impact on cell types involved in gliomas, including neurons, microglia, and astrocytes. Throughout glioma formation, pathogenesis tends to alter these LGICs, inducing specific effects within each cell type that often promote tumorigenesis.

### 2.1. Neurons

Recently, neurons have become important contributors to glioma progression. Neuronal LGICs such as AMPAR, NMDAR, and GABA_A_ have been implicated in tumorigenesis [23,24]. AMPARs are calcium ligand-gated channels that are overexpressed in gliomas, particularly the GluR1 and GluR4 subunits, and play a role in malignancy [23,25]. Studies have shown the pro-tumorigenic effect of AMPARs is potentially caused by their regulation of glutamate-induced calcium oscillations, leading to an overload of extracellular glutamate [23,26]. Glutamate is known to further promote neuronal and oligodendrocyte migration during development, displaying the potential for upregulated AMPAR to indirectly promote glioma migration [27]. Regarding the impact of AMPARs on gliomas, GluR1 expression has been shown to be associated with changes in glioma cell shape [23]. In GluR1 overexpression, staining has shown increased actin polymerization and focal adhesion to type 1 collagen, potentially proving attenuated extracellular matrix (ECM) binding and tumor migration [23].

Interestingly, AMPAR/NMDAR activation typically induces cell death in neurons, but paradoxically in glioma cells, glutamate-receptor activation promotes proliferation and invasion. Recent research has considered that glioma cells downregulate these ligand channels to optimize survival in a glutamate-abundant environment [28]. Uniquely, the majority of AMPARs and NMDARs within gliomas are on cells at the invasive front of the tumor [29].

Like AMPARs, NMDAR ligand channels play a pivotal role in neuronal survival but have also been seen to enhance the growth of gliomas [30]. NMDARs are composed of two GluN1 subunits, two GluN2A-D subunits, and potential GluN3 subunits, which vary depending on the localization within the brain [31]. These channels work via activation of the calcium/calmodulin kinase (CaMK-II/IV) and the mitogen-activated protein (MAPK) pathways leading to the eventual expression of early response genes (ERGs) [32,33]. Studies have shown that NMDAR activation leads to topoisomerase-2-beta (T2B) induced double-stranded breaks in ERGs and that improper repair of these DSBs can induce carcinogenesis and disease [34]. Regarding their implication in gliomas, overexpression of NMDARs is associated with worse prognosis in tumor patients [30,35]. Specifically, the T2B mediated DSBs eventually lead to the expression of proto-oncogenes cFOS and ERG1, which are associated with radioresistance and worse prognosis in gliomas [30]. T2B has also been seen to be overexpressed in glioma-initiating cells, suggesting an alteration of NMDAR activity by gliomas [36,37]. Further, subunit GluN2B has also shown relevance in tumor initiation showing the potential for therapeutic targeting of NMDAR function in the future [35].

### 2.2. Microglia

Microglia are the cells responsible for the CNS immune response. Like neurons, microglia express a variety of LGICs to help communicate with their environment. Particularly, the P2X purinergic receptor and several glutamate receptors play a prominent role in microglial-neuronal crosstalk [38]. It has been shown that GBMs and gliomas show strong upregulation of P2RX7 [39]. Subsequent studies have shown the association between P2RX7 inhibition and the reduction of tumor enlargement within gliomas [40]. P2RX7 confers a unique cell growth advantage via pumping up energy stores due to its ability to activate cation channels at low ATP concentrations [41]. Studies have shown the ability for transfection of P2RX7 to promote growth, potentially by the expression of factors involved in tumor progression and metastasis [40,41,42]. Currently, it is known that P2RX7 has a unique impact on the mitochondrial promotion of growth during tumor progression [42]. However, P2RX7s have been shown to promote the release of pro-inflammatory cytokines, including interleukin (IL)-1β, IL-6, and tumor necrosis factor (TNF)-α, which would limit tumor growth [43]. It is not well known the impact that gliomas have on these ligand-gated channels, but they could potentially include a role in triggering matrix metalloproteinase 2 (MMP-2). Through this mechanism, tumors could evade ATP-induced cytotoxicity and P2RX7 at high concentrations [39]. However, further research is needed on the impact that gliomas have on the P2X ligand channel, as the unique changes that GBMs and other gliomas induce on P2RX7 are a potential therapeutic target for tumor progression.

### 2.3. Astrocytes

Astrocytes, a subset of glial cells, are the most abundant cells in the CNS and are vital in the function of the blood–brain barrier (BBB) [44,45]. Furthermore, astrocytes have been shown to mediate the recycling of neurotransmitters. Historically, astrocytes have been marked as non-excitable but recent research has unveiled their role in cell signaling through response to glutamate [45]. The use of ligand-gated channels such as ionotropic glutamate receptors allows astrocytes to respond to intracellular fluctuations in calcium [46,47]. They also have various ligand-gated calcium channels, which are known to be fundamental to CNS function. Astrocytes’ ability to induce gliosis, the process in which astrocytes repair tissue in response to CNS damage, is also used in response to GBM progression [48,49]. Interestingly, gliosis may contribute to tumor growth. Factors such as nuclear factor kappa-B (Nf-kB) potentially lead to the activation of tumor-associated astrocytes (TAAs) [50]. Further, ligands such as receptor activators of Nf-kB are seen to be produced by GBM cells leading to an eventual increase in TAAs [50]. Further ligands such as TGF-Beta also can promote glioma cell invasion [50]. Similarly, the sonic hedgehog (SHH) pathway portions are seen to be upregulated in gliomas [51]. Issues with signaling regulation of this pathway are associated with the initiation of brain tumors.

Regarding the role of astrocyte ligand channels in gliomas, the expression at the membrane of both tumor cells and astrocytes provides a potential cross-link and further promotes tumor progression [52]. This communication confers the ability of glioma cells to regulate both H+ and calcium concentrations, contributing to the epithelial–mesenchymal transition common in gliomas and leading to further survival of the glioma [52]. Recently, the calcium-regulated release of vesicles has been considered for gliotrasmission [53,54]. This crosstalk between gliomas and astrocytes via released vesicles is seen to be more common, and their ability to transport specific proteins promotes GBM progression. Specifically, chloride intracellular channel proteins (CLICs) are involved in glioma progression [55]. CLIC-1 medicated glioma expansion via secreted extracellular vesicle communication has been recently considered a therapeutic target due to CLIC-1′s upregulation correlating with worse prognosis in GBM patients [56].

## 3. Role of Chemotaxis in the Spread of Gliotic Changes

Chemotaxis, characterized by the migration of cells or organisms in response to an extracellular chemical gradient, plays a crucial role in the progression of gliomas [57]. In the presence of tumors from the brain and spinal cord, chemokine receptors constitute the vast majority of ligand channels frequently engaged in chemotactic events [58]. Chemokine receptors enable the binding of chemokines as a ligand, ultimately serving as a chemoattractant for neighboring glioma cells [59]. Simply put, in response to the chemical messages emitted by damaged cells, glial cells—such as astrocytes and microglia—relocate toward an afflicted injury site (Figure 2) [60]. At the location of interest, glial cells emit additional chemical signals that attract supplemental cells, resulting in an overall increased inflammatory cascade [61]. This mechanism is of utmost interest to neurosurgeons, as the use of chemotaxis by tumor cells can result in secondary brain injuries in cases where the inflammatory response overwhelms surrounding healthy tissue [61]. As such, chemotactic pathways are thought to serve a crucial part in the rapid spread of gliotic changes, attesting to glioma’s invasive nature [57]. With this in mind, the aggressiveness of gliomas abetted by chemotaxis must be further scrutinized to comprehensively understand their prognosis and improve mortality rates amongst affected patients.

### Role of Proteases

To physically infiltrate the blood vasculature, the induction of proteolytic degeneration via proteases is essential [62]. In particular, matrix metalloproteinases, cysteine cathepsins, and serine proteases are a series of proteases that aid in the cleaving of cell-adhesion components, such as epithelial (E)-cadherin, which results in the interference of cell-to-cell junctions [63,64]. Individual or group-mediated tumor cell migration is facilitated by the loosening of these cell connections, where the turnover of proteins in the ECM and basement membrane permits invasive cells to move into the surrounding tissue and vasculature [65]. Proteases are not only essential for the degradation of extracellular proteins, but they also have specialized processing roles that are important for cell signaling. For instance, proteases can activate growth factors and cytokines, which significantly increase chemoattraction, cell migration, and metastasis [66]. These distinct ways of protease-enhanced invasion are not mutually exclusive; rather, they likely work in tandem to increase the spread of cancer cells. All of these activities are closely controlled by a cascade of protease interactions, which allows for the regulation and amplification of proteolysis during the invasion [67]. Therefore, when certain of these protease families are pharmacologically inhibited or genetically abrogated, a significant decrease in cancer cell invasion has been reported [68,69]. Evidently, understanding the processes of chemotaxis in aggressive gliomas might lead to the development of novel therapeutic techniques for treating these tumors.

## 4. Role of Ion Channels in Glioma Cell Signaling

Glioma pathogenesis is influenced by the formation of tumor microtubules which enables direct communication between glioma cells [70]. In addition, various autocrine and paracrine mechanisms have the potential to encourage glioma development and progression. As such, faulty LCIG activity has the ability to significantly impact and alter intracellular signaling pathways implicated in glioma pathogenesis. For example, direct communication between glioma cells and neurons is mediated by functional bona fide chemical synapses [70]. These synapses are located on tumor microtubules and produce signals mediated by AMPA receptors, allowing for calcium-mediated glioma cell invasion and growth [70]. GBM tumors additionally have the ability to form networks mediated by tumor microtubules that allow for significant intratumoral communication [71]. As with low-grade gliomas, glutamatergic signaling mediated by AMPA receptors is associated with increased GBM invasion and proliferation [71]. Specifically, longitudinal time-lapse imaging in vivo revealed GBM transition over time to form interconnected networks that allow for tumor growth and invasion, a process mediated by glutamatergic activation of tumor microtubules [71]. Furthermore, EGFR signaling has the ability to enhance glutamate release from glioma cells [72]. Once activated, EGFR has the potential to phosphorylate GluN2B, a subunit of the NMDA receptor [73]. This results in enhanced NMDA signaling and glioma cell migration [73]. Thus, EGFR signaling plays a crucial role in intracellular communication between glioma cells.

## 5. Role of Genetics in Faulty Ligand Channel Activity

Dysfunctional ligand channel activity is associated with the progression of various tumors, but these alterations are not yet readily identifiable in the clinical setting [74]. Median survival for patients depends on the stage/type of glioma, with GBM having the worst prognosis of only 15 months [75,76]. A possible future direction in determining prognosis is by studying genetic factors related to ligand channel-associated glioma proliferation. One key study by Palmieri et al. [77] led to the discovery of CACNA2D3, a calcium channel gene associated with breast cancer metastasis. Studies such as this show the potential that identifying changes in ion channel genes could have on clinical decision-making. However, isolating single genes may not be the most appropriate strategy as tumor pathogenesis often involves multiple parallel pathways [78]. As a result, the focus of research in ligand channel genes has shifted from individual genes to identifying molecular signatures that combine patterns of expression from multiple genes. These molecular signatures can then be used to predict outcomes for patients with cancer, including those with glioma [79]. Two molecular signatures that have been found to predict survival in glioma patients are a three-gene potassium channel signature and an ion Channel based Gene (iCG) composed of 18 ion channel genes.

Potassium channels promote cell proliferation by contributing to the progression of the cell cycle, and their activity has been linked to tumor development and growth, particularly in GBM [80]. Three genes were found to play an important role in the malignant progression of GBM: KCNN4, KCNB1, and KCNJ10 [80]. In this same study, GBM progression resulted in the upregulation of KCNN4 and the downregulation of KCNB1 and KCNJ10 [80]. This three-ion channel molecular signature was used to develop a risk score formula for patients with GBM to predict their overall survival rates. For example, a cohort of Chinese patients was divided into high-risk and low-risk groups based on the presence of this molecular signature, and their survival times were compared [80]. In this study, the high-risk group had a shorter median overall survival when compared to the low-risk group across each data set assessed [80]. Data from the Chinese Glioma Genome Atlas, Cancer Genome Atlas, and REMBRANDT were compiled [80]. Each data set was statistically significant, suggesting that the three ion channel molecular signature can be used to predict the prognosis of primary GBM patients [80]. Another study observed 18 ion channel genes (Table 2) in patients with varying grades of glioma to create the iCG molecular signature [5]. Grades I-II of glioma were classified as low-grade patients, and grades III-IV of glioma were classified as high-grade patients [81]. Of the 18 genes, 16 were found to be down-regulated (−1) in high-grade glioma, while 2 were up-regulated (1). The iCG signature included calcium, chloride, potassium, and sodium channels.

Subsequent analysis of this iCG signature was conducted in high-grade glioma patients from the European Organisation for Research and Treatment of Cancer (n = 95), M.D. Anderson Cancer Center (n = 77), and Massachusetts General Hospital (n = 50) [5]. The presence of the iCG marker predicted a significantly higher mortality risk when compared to controls [5]. Of note, the *GLRB, GRIA2, GRID1,* and *P2RX7* genes code for ligand-gated ion channels discussed in this review. This further supports the prognostic value of determining LGIC expression for glioma prognosis.

The three gene and iCG signatures are the only two known examples showing associations between ligand channel genetics and clinical outcomes in patients with glioma. The genetics behind ligand channels could become a new, effective way to predict glioma outcomes while strengthening our understanding of the links between LGIC overexpression and glioma pathogenesis [82]. Further research could advance this field of study by finding other molecular signatures that can be used to measure glioma progression.

## 6. Overview of PANX2 Channels

Examining biomolecular changes in glioma patients has led many scientists to investigate the role of PANX2 channels in gliomas [83,84]. PANX2 channels contain glycoproteins and form single membrane channels between cells in the central nervous system, particularly in astrocytes and microglia [85]. Their membrane topology is similar to connexin hemichannels, but PANX2 is often shown to be in heptamers and octomers [86]. They are unable to form proper gap junctions between cells when compared to connexins despite sharing similar sequence homologies. Due to PANX2 missing two cysteine residues [85,87], its channel is unable to adhere properly to other cells and transfer large molecules between neuronal cells [88]. However, more recent studies using expansive technology have shown tiny vesicles branching from PANX2 in proximity to actin filaments, leading to the idea of the transfer of much smaller molecules [89].

PANX2 also has functions in cell differentiation, inflammation, tissue remodeling, and wound healing (Figure 3) [90,91]. PANX2 mRNA is restricted primarily to the cerebral cortex, cerebellum, temporal lobe, and medulla, suggesting PANX2 plays an important role in neuronal cell differentiation and communication [86,92]. PANX2 are more commonly found within the neuronal cells in their cytoplasm or within membrane-bound organelles, such as the mitochondria [93]. Although not highly researched, studies have shown that the role of PANX2 in ATP regulation during cancers might connect with the purinergic role of ATP in various neurological disorders [94]. Purinergic receptors have often functioned in neuroinflammatory responses. Rapid excretion of ATP from cancerous cells has been seen to further activate nearby microglial cells. Lohman et al. [88] used various clinical studies to examine the role of PANX2′s cellular release of ATP in regulating inflammation within the brain. Inflammatory responses and unregulated inflammation have been shown to promote glioma growth [88,95].

One experimental study analyzed the connection between PANX2 and cancer immune infiltrates using a technique called Tumor Immune Estimate Resource using NIH’s data portal [22]. Their research database showcased an increase in levels of PANX2 and correlated it with increased survival and better outcomes for glioma patients. After analyzing numerous clinical variables such as tumor type and radiation treatment, a model that predicted the possible outcome of a patient was created based on the varying levels of PANX2 in patients [22]. Significant reductions in PANX2 levels were reported, but subsequent analysis is still being conducted on this model [22]. Another team of researchers reported decreased or absent levels of PANX2 mRNA in cultured human glioma cells [93]. When compared to PANX2′s involvement in oncogenicity, High Throughput cDNA microanalysis showed that PANX2 acted similarly to tumor suppressor genes due to the reduction of tumor cells when PANX2 was restored to normal levels [93]. The use of anti-PANX2 antibodies showed negligible or limited PANX2 expression in human glioma cells [93].

The search for targeted therapeutic strategies or genetic therapeutic approaches to battling high-grade gliomas is vastly important. Despite this extensive research and clinical analysis, PANX2 has a void of uncertainty. New studies need to be implemented to examine its potential to provide care and treatments for patients battling life-threatening gliomas [96].

## 7. LGICs as a Clinical Target

Advances in understanding the role of LCIGs in glioma pathogenesis have led to the investigation of these channels as a potential glioma therapeutic. LGIC-targeted pharmacotherapies mainly focus on altering LCIG ligand concentrations. At this point, trials largely remain in the preclinical phase.

### 7.1. Purinoreceptors

The P2RX7 has been the most studied of the purinoreceptors in regard to cancer proliferation and thus appears to be the most promising avenue for therapeutics [43]. Interestingly, activation of P2RX7s can both inhibit and promote tumor growth depending on the level of receptor activation. At median levels of activation, the P2RX7 induces cell proliferation. At high levels of extracellular ATP, P2RX7 mediates caspase-3 cleavage with subsequent membrane degradation and cell death [43,97]. As mentioned previously, P2RX7 activation is thought to promote immune response through the release of pro-inflammatory cytokines. This spectrum of activation has been therapeutically leveraged to treat glioma. For example, Douguet et al. [98] developed a P2RX7 agonist, HEI3090, which sensitized non-small cell lung cancer to immunotherapy, inducing tumor regression in 80% of cancer-bearing mice.

Aside from reducing tumor burden, targeting purinoreceptors is a new avenue for mitigating cancer comorbidities. Other P2XR subtypes have been associated with cancer-related symptoms. One salient example is neuropathic and inflammatory pain secondary to cancer growth and inflammatory state, respectively. Specifically, the P2RX2 and P2RX3 receptor subtypes expressed on afferent neurons have been shown to mediate cancer-associated pain [94,99]. Using a P2RX3 and P2RX2/3 receptor antagonist, AF-353, Kaan et al. [100] were able to decrease bone cancer pain in a rat model.

### 7.2. Glutamate-Gated Receptors

The NMDA and AMPA glutamate receptors are amongst the most studied glutamate-gated receptors for their role in tumorigenesis [101,102]. There are anti-epileptic medications that have mechanisms of action at the site of the glutamate receptor that, in recent years, have been studied for their potential role in glioma treatment [103]. For example, Perampanel is a non-competitive AMPA glutamate receptor antagonist that is FDA-approved to treat partial onset seizures. In GBM cell lines, Perampanel has been shown to have antiproliferative effects via a decrease in glucose uptake, slowing cell metabolism [104]. Talampanel is an allosteric inhibitor of the AMPA receptor that is currently being studied for its role in treating seizures, ALS, Parkinson’s disease, and GBM. Vigabatrin is an irreversible GABA transaminase inhibitor used to treat refractory seizures and infantile spasms that are being studied for treating brain tumors and, more specifically, gliomas [105]. Beyond the treatment of gliomas, glutamate-gated receptors have been implicated in other disease processes and have varying clinical trials (Table 3). Riluzole is a medication FDA-approved for the treatment of amyotrophic lateral sclerosis (ALS). Its mechanism of action is not well-established but is thought to involve the inhibition of voltage-gated ion channel release of glutamate [106]. AMPA receptor activation has been shown to increase pancreatic cancer cell invasion [107]. North et al. showed that human ovarian and small-cell lung cancer lines express NMDA receptors and are potential targets for treatment [108,109].

### 7.3. Cys-Loop Receptors

Like the glutamate-gated receptor superfamily, Cys-loop receptor agonists/antagonists are widely FDA-approved for other neurological diseases. One historical link is between benzodiazepines, which are GABA_A_ agonists, and the proliferation of breast cancer. In this study from over 30 years ago, Kleinerman et al. [110] observed that patients on diazepam exhibited smaller, less invasive tumors with less lymph node involvement compared to patients not on the medication. Findings such as this have prompted other researchers to further explore GABA agonists as potential cancer therapeutics.

Valproic acid (VPA) is a commonly used anti-epileptic that has been repositioned in recent years to treat both adult and pediatric glioma (NCT00879437, NCT00302159). VPA’s mechanism of action is mainly through the increase of presynaptic GABA, but recent research has suggested that VPA is a histone deacetylase inhibitor that alters the acetylation pathways in human gliomas [111,112,113]. This mechanism is similar to the already commonly used anti-brain tumor therapy, TMZ. VPA has been studied in animals and cell lines as a radiosensitizer for glioma. It has been shown that VPA-enhanced radiation-induced cell death in the C6 rat glioma cell line [114,115]. Su et al. found that VPA use as a radiosensitizer was well-tolerated in pediatric DIPG but ultimately did not result in an overall survival benefit [116].

As with our discussion of purinoreceptors, Cys-loop receptors have been leveraged to alleviate cancer-associated symptoms. Therefore, this relationship could be therapeutically leveraged to improve the quality of life for patients with brain tumors. For example, LGICs have also been associated with the development of glioma-related epilepsy (GRE) in diffuse glioma patients [117]. As such, GABA_A_ modulators have been commonly used in both general epilepsy and GRE. In addition to its potential use in decreasing glioma proliferation, VPA has also been studied in regard to GRE. NCT03048084 is currently recruiting patients to see which anti-epileptic is most efficacious in GRE (NCT03048084). The combinatorial benefit of seizure prophylaxis and tumor suppression would make VPA a very potent therapeutic for patients with brain tumors.

## 8. Conclusions

It is well established throughout the literature that glioma pathogenesis has the potential to significantly modulate the expression and structure of LGICs. Pathological changes to LGICs can exacerbate glioma progression, tumor heterogeneity, and patient symptomatology. This phenomenon has led to the investigation of LGICs as a potential diagnostic tool and therapeutic option for gliomas. However, it is important to note that many LGIC-based trials remain in the preclinical phase highlighting the need to refine current and emerging LGIC-based glioma diagnostic and therapeutic modalities.

One example of an emerging LGIC-based therapeutic for glioma involves the administration of capsaicin, a neurotoxin widely found in chili peppers. When ingested, capsaicin produces a ‘spicy’ sensation through the activation of vanilloid type-I (TRPV1) receptors [118]. TRPV1 activation increases calcium ion influx resulting in an increased cytosolic concentration of calcium, which can lead to apoptosis [119]. Consequently, capsaicin has been investigated as a potential anticancer therapeutic with increasing evidence demonstrating its efficacy for prostate cancer, glioma, [120] GBM, [118] renal cell carcinoma, [120] and human urothelial cancer [121]. In glioma specifically, subtoxic doses of capsaicin effectively induce apoptosis through the tumor necrosis factor-related apoptosis-inducing ligand (TRAIL) [120]. Additionally, gliomas treated with capsaicin show significant upregulation of TRAIL death receptor 5 mediated by C/EBP homologous protein, also known as growth arrest and DNA damage-inducible gene 153 activation [120]. In this same study, downregulation of the caspase survivin protein was observed, additionally promoting apoptosis [120]. Capsaicin additionally demonstrates anticancer efficacy in more severe primary brain malignancies such as GBM. When capsaicin is administered to human GBM LN-18 cells, peroxisome proliferator-activated receptor gamma is expressed, facilitating GBM tumor cell apoptosis [118].

One of the major challenges associated with the effective treatment of glioma is the BBB, a network of tight junctions that permit the passage of molecules and drugs into the brain in a highly selective manner. One way to overcome the BBB is to deliver therapeutics enclosed in lipid-soluble nanoparticles. A variety of nanocarriers, including liposomes, micelles, inorganic systems, polymeric nanoparticles, nanogels, biomimetic nanoparticles, and prodrug conjugate self-assembled nanoparticles, have been explored for the delivery of glioma drugs [122]. When modified by the addition of glioma-specific ligands, liposomes specifically are popular nanoformulations as they are easy to produce, express high compatibility and biodegradability, and can release drugs in a highly controlled manner [122]. In glioma, liposomes have been shown to effectively deliver drugs to the site of the tumor to facilitate an immune response [123,124,125,126,127] and remodel the tumor microenvironment [128,129], which becomes highly immunosuppressive, resistant to standard therapeutics, and demonstrate angiogenesis in glioma. Although there are no reports in the literature detailing LGIC liposome-based drug delivery strategies, it could be proposed that liposomes and other nanoparticles could enhance the delivery of LGIC-based glioma therapeutics.

Because genetics play a significant role in the pathogenesis of glioma, molecular sequencing is important to identify potential therapeutic targets. With the advancement of molecular subtyping in glioma diagnosis per the 2021 WHO guidelines, we suspect that genetic alterations in the aforementioned ionotropic channels may soon become clinically significant. Therefore, it is possible that the development and refinement of LGIC-based therapeutics may be facilitated by the identification of patient-specific glioma subtypes.

In sum, although significant progress has been made in understanding the role of LGICs in glioma pathogenesis, future investigations are still needed to refine LGIC-based diagnostic tools and therapeutics before they are implemented into clinical practice.

## Figures and Tables

**Figure 1 jpm-13-00853-f001:**
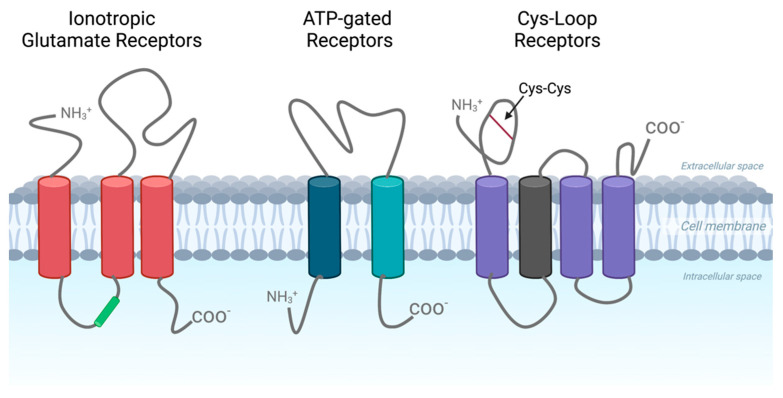
**Ligand-gated Ion Channel Classifications.** Ionotropic Glutamate receptors, ATP-gated receptors, and Cys-Loop Receptors. This figure illustrates the receptor’s conformational changes (open state) and post-activation by their respective ligands. Figure created with BioRender.com.

**Figure 2 jpm-13-00853-f002:**
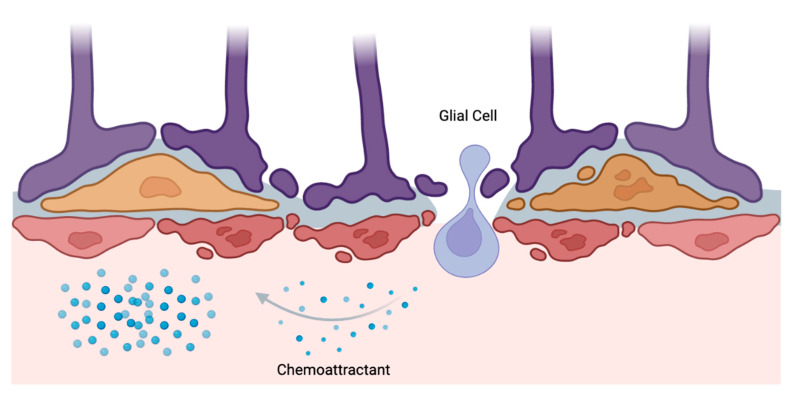
Illustrative depiction of chemotaxis in gliomas. A glial cell can be seen undergoing intravasation to the site of injury.

**Figure 3 jpm-13-00853-f003:**
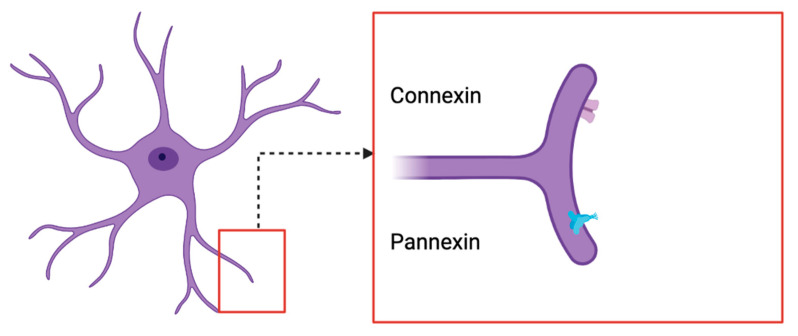
In this image, we can visualize the similarities and differences in structure between connexins and pannexins. In terms of channel morphology, pannexins do not form large channels for distribution of molecules and ions such as connexins. Take note that due to the two extra cysteine residues in connexin molecules, they produce a larger channel [85,87]. Due to this variability in size, connexins are better able to transfer ions and molecules between neurons. Pannexin vesicles often branch from their actin filaments and participate in the transfer of much smaller molecules [89].

**Table 1 jpm-13-00853-t001:** Subtypes of Ligand-gated Ion Channels [10].

Receptor	Receptor Subtype	Ligand	Ionic Conductance
Ionotropic Glutamate	AMPA	L-Glutamate	Na^+^, K^+^, Ca^2+^
	Kainate	L-Glutamate	Na^+^, K^+^
	NMDA	L-Glutamate	Na^+^, K^+^, Ca^2+^
ATP-gated channel	P2X	ATP	Na^+^, K^+^, Ca^2+^
Cys-Loop	nACh	Acetylcholine	Na^+^, K^+^, Ca^2+^
	5-HT_3_	Serotonin	Na^+^, K^+^
	GABA_A_	GABA	Cl^−^, HCO_3_^−^
	Glycine	Glycine	Cl^−^

**Table 2 jpm-13-00853-t002:** Genes involved in the pathogenesis of glioma.

Gene	Gene Description	Weight
*CACNA1D*	calcium channel, voltage-dependent, L type, alpha 1D subunit	−1
*CLCN6*	chloride channel, voltage-sensitive 6	−1
*CLIC1*	chloride intracellular channel 1	1
*CLIC4*	chloride intracellular channel 4	1
*GLRB*	glycine receptor, beta	−1
*GRIA2*	glutamate receptor, ionotropic, AMPA 2	−1
*GRID1*	glutamate receptor, ionotropic, delta 1	−1
*KCNAB1*	potassium voltage-gated channel, shaker-related subfamily, beta member 1	−1
*KCNB1*	potassium voltage-gated channel, Shab-related subfamily, member 1	−1
*KCND2*	potassium voltage-gated channel, Shal-related subfamily, member 2	−1
*KCNJ10*	potassium inwardly-rectifying channel, subfamily J, member 10	−1
*KCNMA1*	potassium large conductance calcium-activated channel, subfamily M, alpha member 1	−1
*KCNN3*	potassium intermediate/small conductance calcium-activated channel, subfamily N, member 3	−1
*KCNQ5*	potassium voltage-gated channel, KQT-like subfamily, member 5	−1
*NALCN*	sodium leak channel, non-selective	−1
*P2RX7*	purinergic receptor P2X, ligand-gated ion channel, 7	−1
*SCN1A*	sodium channel, voltage-gated, type I, alpha subunit	−1
*VDAC2*	voltage-dependent anion channel 2	−1

**Table 3 jpm-13-00853-t003:** Registered clinical trials and their current status.

Study	Title	Status	Receptor Type	Conditions
NCT02082821 (Boston, MA, USA)	A P2X7R Single Nucleotide Mutation Promotes Chronic Allograft Vasculopathy	Completed	P2RX7	Cardiac Allograft Vasculopathy
NCT03918616 (Pisa, Italy)	P2X7 Receptor, Inflammation, and Neurodegenerative Diseases	Completed	P2RX7	Neuro-Degenerative Disease
NCT02293811 (Alpes-Maritimes, France)	Decoding of the Expression of Tumor Suppressor P2RX7 in Inflammatory and Malignant Colonic Mucosa (P2RX7)	Unknown	P2RX7	Crohn’s Disease-Associated Colorectal Adenocarcinoma
NCT04122937 (Pisa, Italy)	Defining Inflammation Related to Peritoneal Carcinomatosis in Women With Ovarian or Colon Cancer (CarFlog)	Completed	P2RX7	Peritoneal Carcinomatosis Ovarian Cancer Colon Cancer
NCT05225883 (Lyon, France)	GWAS in NMDAR Encephalitis	Recruiting	Glutamate NMDA	Autoimmune Encephalitis
NCT05503264 (Birmingham, AL, USA)	A Study To Evaluate The Efficacy, Safety, Pharmacokinetics, And Pharmacodynamics Of Satralizumab In Patients With Anti-N-Methyl-D-Aspartic Acid Receptor (NMDAR) Or Anti-Leucine-Rich Glioma-Inactivated 1 (LGI1) Encephalitis (Cielo)	Recruiting	Glutamate NMDA	NMDAR Autoimmune Encephalitis LGI1 Autoimmune Encephalitis
NCT02654964 (Washington, D.C., MA, USA)	Cancer Stem Cell High-Throughput Drug Screening Study	Recruiting	Glutamate NMDA	GBM
NCT02363933 (Durham, NC, USA)	Perampanel in Seizure Patients With Primary Glial Brain Tumors	Completed	Glutamate AMPA	Brain Tumor, Primary
NCT03636958 (Paca, France; Marseille, France)	Efficacy and Safety of Perampanel in Combination in Glioma-refractory Epilepsy	Withdrawn	Glutamate AMPA	Refractory Epilepsy
NCT03229278 (New Brunswick, NJ, USA)	Trigriluzole with Nivolumab and Pembrolizumab in Treating Patients with Metastatic of Unresectable Solid Malignancies or Lymphoma	Completed	Cys-loop	Lymphoma Unresectable or metastatic malignancies
NCT00879437 (Dallas, Fort Worth, Houston, and San Antonio TX, USA; Oklahoma City, OK, USA)	Valproic acid, radiation, and bevacizumab in children with high-grade gliomas or diffuse intrinsic pontine glioma	Completed	Cys-loop	Pediatric high-grade glioma or brainstem glioma
NCT00302159 (Bethesda, MA, USA; Philadelphia, PA, USA; Richmond, VA, USA)	Valproic acid with temozolomide and radiation therapy to treat brain tumors	Completed	Cys-loop	GBM High-grade giomas
NCT03048084 (The Hague, Netherlands; Rotterdand, Netherlands; Amsterdam, Netherlands)	Seizure treatment in glioma (STING)	Recruiting	Cys-loop	Glioma

Available online: Clinicaltrails.gov (accessed on 2 May 2023).

## Data Availability

Not applicable.

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
