# Peer review of "Ligand-Gated Ion Channels: Prognostic and Therapeutic Implications for Gliomas"

_jpm, 2023, doi:10.3390/jpm13050853_

Round 1
Reviewer 1 Report
This is a very complete review on an original subject that is still relatively unknown, especially to pathologists, but is growing rapidly. This review is very well written. The tables are useful and the figures of very good quality.
I have only one minor suggestion in the introduction concerning the description of the WHO 2021 classification: I would take another example than TSC in SEGA which are still very rare tumors (e.g. IDH in diffuse glioma of adult type)
Author Response
Point 1: Suggested to use a different example of genetic/molecular biomarkers used by the WHO to classify CNS tumors than TSC in SEGA.
Response 1: We agree with reviewer 1 that a more common example should be cited in this review. As such, we have replaced the original sentence with genes and biomarkers used by the WHO to classify glioblastoma.
Reviewer 2 Report
Dear Authors,
Your manuscript on the role of Ligand-Gated Ion Channels in glioma prognosis and therapy is a valuable contribution to the field of glioma research. I do agree that this is an underrepresented topic with high potential for future therapeutic approaches. Nevertheless, there are some points that should be addressed to allow a better understanding of your article.
1. You use several ways of writing of the purinergic receptor P2X7. To avoid confusion I would recommend to stick to one way of writing. In addition I recommend to mention that there are several ways of writing. Especially P2RX7 should be mentioned as this is the way it is deposited in most databases.
2. Please be very carefully when citing literature and include the data presented in your manuscript. There are several instances in which your description differs from the literature cited; e.g. line 308-311, neither western blot nor immunofluorescence can be used to show mRNA levels; line 339, “minimal antibodies” is not the correct wording for this data; Line 340-343 mentions a clinical study which is not part of your citation 93.
3. Why is paragraph 3.2 included? This is no ion channel, so it makes no sense to me to include it in this review. I would recommend to explain the inclusion a bit more or leave it out.
4. I would recommend to mention that the interconnection of glioma cells and their signaling is a topic that gets high attention recently (PMID: 35914528, PMID: 31534219, …), as this is also based on ion channels.
Minor remarks:
Line 75-78 is hard to understand. How is FAK related to the glutamine?
Figure 4 is misleading. The channels reside in the membrane and connect the cytoplasm with the extracellular space. In this figure it looks like they go through the whole cell spanning two membranes.
Line 353 Do you really speak about “repositioning” of drugs? I rather think this should be “repurposing”.
Line 390-393 look a bit stretched in my version. I think it should be closer together.
Line 411, it is essential to mention that C6 is a rat glioma cell line and no human. It is a nice glioma model but genetically far away from the glioma it should represent.
Best regards
Author Response
Point 1: Suggested to use one abbreviation for the purinergic receptor P2X7.
Response 1: We agree with reviewer 2 that one abbreviation should be used throughout the manuscript for clarity. As such, all abbreviations of the purinergic receptor P2X7 have been changed to “P2RX7” as this is how it is deposited in most databases.
Point 2: Suggested to carefully check citations to ensure what is written in the manuscript aligns with the citation. Additionally, this point suggested to double check data presented in the review for scientific accuracy.
Response 2: We agree with reviewer 2 regarding these comments. Lines 308-311 (in the most updated manuscript, this is lines 377-379) have been modified to highlight specific areas of the brain PANX2 plays a role in rather than stating immunofluorescence and western blotting can be used to show mRNA levels. Additionally, in line 339 (in the most updated manuscript, this is lines 415-416) rather than saying “minimal antibodies”, this has been modified to say, “negligible or limited PANX2 expression”.
Point 3: Suggested to remove paragraph 3.2 because it does not describe an ion channel.
Response 3: We agree with reviewer 2 that paragraph 3.2 should be removed because it talks primarily about the role of CXCR4 and CXCL12, both of which are not ion channels. Because this paragraph is beyond the scope of our review, we have decided to remove it.
Point 4: Recommends highlighting the role of ion channels glioma cell signaling.
Response 4: We agree with reviewer 2 that this is an important aspect to include in this review. As such, we have included section 4. Role of Ion Channels in Glioma Cell Signaling.
Point 5: Stated that lines 75-78 were difficult to understand and suggested to elaborate further for clarity.
Response 5: We agree with reviewer 2 that lines 75-78 (in the current manuscript, this is lines 78-81) were unclear. We have clarified this sentence in the most updated version of the manuscript.
Point 6: Suggested that figure 4 is misleading because the channels appear to span the entire cell rather than connecting the cytoplasm with the extracellular space.
Response 6: We agree with reviewer 2 that figure 4 (in the most updated manuscript, this is figure 3) should be modified. As such, we have modified the figure to show the channels connecting the cytoplasm with the extracellular space rather than spanning the entire cell.
Point 7: Suggested to reword “repositioning” in line 353 to say “repurposing.”
Response 7: For clarity, this entire paragraph (lines 423-426) has been reworded.
Point 8: Suggested to make lines 390-393 closer together.
Response 8: This appears to have been a formatting issue. Lines 390-393 (now 514-515) have been edited to remove the excess space.
Point 9: Suggested to mention C6 is a rat glioma cell line.
Response 9: We agree that it is critical to highlight C6 is a rat glioma model. As such, we have indicated that in the manuscript in line 544.
Round 2
Reviewer 2 Report
Dear Authors,
Thank you for addressing all remarks so fast and thoroughly.
Best regards.